# Synthesis of I(III)/S(VI) reagents and their reactivity in photochemical cycloaddition reactions with unsaturated bonds

Li Li[1], Kun Deng[2], Yajie Xing[1], Cheng Ma[3], Shao-Fei Ni[3], Zhaofeng Wang [2] ✉ & Yong Huang [1] ✉

The development of novel methodologies for the introduction of the sulfoxonium group under mild conditions is appealing but remains underexplored. Herein we report the synthesis of a class of hypervalent iodine reagents with a transferrable sulfoxonium group. These compounds enable mixed iodonium-sulfoxonium ylide reactivity. These well-defined reagents are examined in visible-light-promoted cyclization reactions with a wide range of unsaturated bonds including alkenes, alkynes, nitriles, and allenes. Two distinct cyclization pathways are identified, which are controlled by the substituent of the unsaturated bond. The cycloaddition protocol features simple operation, mild reaction conditions, and excellent functional group tolerance, affording a broad range of sulfoxonium-containing cyclic structures in moderate to excellent yields. Furthermore, the sufoxonium group in the product can be transformed into diverse functional groups and structural motifs via single electron transfer and transition-metal catalysis.

Hypervalent sulfur-containing compounds are defined as molecules in which the central sulfur atoms are bonded to more than four pairs of valence electrons[1]. During the past few decades, studies of hypervalent sulfur compounds have undergone tremendous growth owing to their biological relevance and chemical versatility[2–6]. There has been a remarkable renewal of interest in molecular design, synthesis, and reactivity exploration of hypervalent sulfur compounds, such as sulfonamides[7,8] and sulfones[9–11], etc. In contrast, sulfoxonium, a class of S(VI) compounds with a S = O double bond and three attached carbons remain underexplored. Compared to its S(IV) analogs, sulfonium salts[12–16], sulfoxonium compounds exhibit higher nucleofugality, increased stability, and decreased basicity. Among them, sulfoxonium ylides have been explored as a diazo-free carbene precursor in synthesis[17–21]. It has been established that sulfoxonium ylides readily form metal-carbenoids in the presence of transition metals and undergo X–H insertion[22–29] and C–H activation reactions[30–34]. Recently, the groups of Burtoloso[35] and Sun[36] demonstrated that sulfoxonium

ylides could be protonated using a chiral Brønsted acid and subsequently undergo enantioselective displacement to form chiral C–S and C–N bonds (Fig. 1a). Despite these advances, exploration of sulfoxonium ylides and their derivatives has been very limited, largely due to the narrow scope of accessible sulfoxonium compounds. The most common method to prepare sulfoxonium salts is the nucleophilic displacement of electrophilic halides with dimethyloxosulfonium methylide (the Corey-Chaykovsky reagent, Fig. 1b)[37]. Monofunctional sulfoxonium ylides could be further branched by reacting with arynes[38] or (hetero)aryl electrophiles[39,40]. An alternative approach applies insertion reactions of sulfoxides into active methylene compounds[41] or carbenes[42–46], which could be generated by thermal, photonic, or transition-metal decomposition of diazo compounds[42–44] or iodonium ylides[45,46]. Notwithstanding the development, the structural diversity of known sulfoxonium ylides remains very limited. Installation of an additional functional group at the ylide carbon is particularly challenging. In this work, we report a general method to

[1]Department of Chemistry, The Hong Kong University of Science and Technology, Clear Water Bay, Kowloon, Hong Kong SAR, PR China. [2]State Key Laboratory of Chemo/Biosensing and Chemometrics, College of Chemistry and Chemical Engineering, Hunan University, Changsha, Hunan 410082, PR China. [3]Department of Chemistry and Key Laboratory for Preparation and Application of Ordered Structural Materials of Guangdong Province, Shantou University, Shantou 515063 Guangdong, PR China. ✉e-mail: zfwangchem@hnu.edu.cn; yonghuang@ust.hk

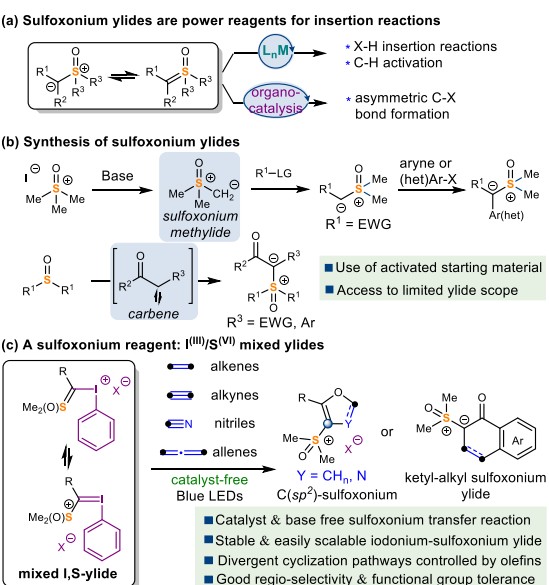

**Fig. 1 | Synthesis and reactivity of sulfoxonium ylides. a** Sulfoxonium reactivities. **b** Previous synthesis of sulfoxonium (ylide). **c** Our strategy: I[(III)]/S[(VI)] mixed ylides as a sulfoxonium reagent.

prepare an I[(III)]/S[(VI)] reagent and describe its reactivity towards unsaturated bonds.

## Results

### Synthesis of the I[(III)]/S[(VI)] reagent

In sulfoxonium ylide-mediated insertion reactions, the sulfoxonium group inevitably falls off to give the key carbene intermediate. To access structurally complex sulfoxonium compounds, we decided to develop a sulfoxonium transfer process using an I[(III)]/S[(VI)] reagent. Hypervalent iodine[(III)] reagents have attracted widespread interest in the synthetic community owing to their capability of transferring functional groups of the same oxidation state[47,48]. In this context, several examples of defined hypervalent iodine reagents containing transferable hypervalent functional groups such as phosphonium[49–51] and arsonium groups[52,53] have been successfully developed for use in single-electron transfer (SET) or electrophilic substitution reactions. Among them, sulfur-containing hypervalent iodine reagents remains scarcely investigated and only one sulfonium variant has been reported by Podrugina and co-workers[54]. However, this S[(IV)] embedded structure was reported to be highly unstable above −10 °C and unable to be isolated. Little was known of its reactivity. We envisaged that a sulfoxonium substitution with a higher S oxidation state might stabilize the negative charge of ylide carbon, offering both stability and new reactivity of I, S-mixed ylides. In this context, we report the design, synthesis, and X-ray structural characterization of I[(III)]/S[(VI)] reagents and their use in photochemical cycloaddition reactions with various unsaturated bonds to afford [3 + 2] or [4 + 2] products containing sulfoxonium functionality (Fig. 1c).

Our initial efforts focused on the synthesis of the I[(III)]/S[(VI)] reagent **3** (Fig. 2). Concerning the stability of **3**, we decided to first synthesize cyclic analogs with a stabilizing *ortho*-carboxylate group, a trick often applied to I[(III)] compounds[47]. 1-Acetoxy-1,2-benziodoxol-3(*1H*)-one **1a** underwent smooth ligand exchange with α-ketosulfoxonium ylides **2** to give the desired products **3a** and **3b** in moderate yield using pyridine and trimethylsilyl triflate (Fig. 2). These compounds are white solids and have reasonable stability in either the solid state or in solution at ambient temperature over an extended time. The corresponding acyclic analogs were prepared from diacetoxyiodobenzene **1b** using a strong Brønsted acid. Excellent yields were obtained on gram scales (see Supplementary Methods for more details). These

acyclic reagents (**3c-3w**) are relatively less stable compared to **3a** and **3b**. They could be stored at −20 °C for several weeks, but gradually decompose at room temperature in a solid state. The counterion of these iodonium salts affects stability, wherein hexafluorophosphates **3c** and tetrafluoroborates **3e** are more stable than triflates **3d**. Versatile substituted aryl- and heteroaryl- derivatives were prepared (**3f-s**), as well as alkyl substituted analogs (**3t, u**) using a set of uniform conditions.

The structures of both cyclic (neutral, **3b**) and acyclic (cationic, **3e**) reagents were unambiguously established by a single-crystal X-ray diffraction analysis (Fig. 2). A closer examination revealed that both **3b** and **3e** adopt the typical T-shaped geometry of a hypervalent iodine atom[47]. For **3e**, the lengths of both C-I bonds are similar to those in a diaryliodonium salt (2.0–2.1 Å), with a C1-I-C11 bond angle of 100.61(1)°. The distance between the iodine and the nearest fluorine atom on the tetrafluoroborate anion (I • • • FBF$_3$), is 2.965(3) Å with a C11-I1-F2 angle of 178.90(1) °, suggesting a type II halogen bonding contact[55]. The shortest S • • • F distance is 3.802(3) Å, indicating the absence of secondary bonding between the sulfur atom and the anion[56]. Partial double-bond connectivity between C1 and C2 (C1-C2 = 1.429(6) Å) was noticed. The coplanar positioning of S1, C1, C2, and O1 atoms (torsion angle = −2.33(5)°) supports the presence of a delocalized S-ylide functionality (S1-C1 = 1.716(3) Å).

### Reaction optimization

We next turned our attention to exploring the reactivity of the synthesized I[(III)]/S[(VI)] reagents. Considering the high leaving group ability of I[(III)][57], we envisioned that these molecules might be activated under photochemical conditions, enabling a sulfoxonium transfer to common chemical feedstocks via a sulfoxonium-substituted reactive intermediate. Inspired by the early work of Murphy and co-workers that employed blue LED (light emitting diode) to promote cyclopropanation of iodonium ylides with alkenes[58], we commenced to explore the chemical reactivity of **3c** towards 4.0 equiv. of 1-hexene in MeCN under blue LED (1 W, λ$_{max}$452 nm) irradiation. To our delight, the photocycloaddition occurred smoothly to afford [4 + 2] product **5c** in 95% isolated yield (Table 1, entry 1). To the best of our knowledge, this benzo-fused cyclic compound is a rare example of ketyl alkyl substituted sulfoxonium ylide. The counter ion of acyclic reagents (**3d** and **3e**) had a moderate effect on the overall conversion (entries 2 and 3). No desired product was detected using the neutral cyclic reagent **3a**, likely due to the lower tendency for neutral I[(III)] to be excited under visible light (entry 4). Solvent screening indicated that halogenated solvents such as DCM decreased reaction efficiency (entry 7). When MeOH was used, neither **5c** or O-H insertion products was detected (entry 6). The [4 + 2] cycloaddition becomes sluggish under aerobic condition and no product was formed in the dark (entry 7 and 8). Using 2 equiv. of 1-hexene and irradiating the reaction mixture with white LED lead to slightly lower yields, and very little improvement was realized when prolonging the reaction time to 5 h (entries 9–11).

### Examination of substrate scope

Having the optimized conditions in hand, we turned our attention to the substrate scope and limitations of the light-mediated tandem [4 + 2] cycloaddition of unactivated alkenes feedstocks **4** with reagent **3**, and the results are summarized in Fig. 3. We were glad to find that this method was able to convert alkene feedstocks containing different straight-chain alkyl substrates into cyclic sulfoxonium ylides, with excellent chemoselectivity (**5a-g**). The functional group tolerance of this transformation is illustrated using substrates bearing cycloalkanes (**5 h**), aromatic rings (**5i, j**), ethers (**5k**), esters (**5l, m**), sulfonates (**5n**), sulfones (**5o**), and silanes (**5p**). The regioselectivity of this reaction was confirmed by converting product **5i** to a known product (see Supplementary Methods for more details). Furthermore, directing groups were not needed to ensure site-selectivity for a substrate bearing two

**Fig. 2 | Synthesis, scope, and X-ray structures of I^(III)/S^(VI) reagents.** X-ray crystal structures of reagents 3b and 3e. The thermal ellipsoids are drawn at 50% probability. **3b:** selected torsion: C23-C22-I1-C1 = −5.77(2)°; C22-I1-C1-C2 = 92.69(2)°. Select bond lengths: I-C1 = 2.059(3) Å, I-C22 = 2.130(2) Å, C1-S1 = 1.715(3) Å, C1-C2 = 1.431(3) Å, O2-C2 = 1.230(4) Å. Selected bond angles: C1-I-C22 = 97.35(1)°, C22-

I1-O3 = 73.88(9)°. **3e:** selected torsion: C1-I1-C11-C12 = −64.99(3)°; C11-I1-C1-F2 = 179.43(1) °. Select bond lengths: I1-C1 = 2.054(4) Å, I1-C11 = 2.111(4) Å, C1-S1 = 1.716(3) Å, C1-C2 = 1.429(6) Å, O1-C2 = 1.248(5) Å. Selected bond angles: C1-I1-C11 = 100.61(1)°, C1-I1-F2 = 80.33(1)°.

reactive alkenes (**5q**). Only the electron-rich terminal alkene was selectively cyclized. *E/Z*-mixtures of **4r** (*E/Z* = 50:50) and **4s** (*E/Z* = 63:37) resulted in the exclusive formation of *trans*-substituted **5r** and **5s**, respectively. These results demonstrated the feasibility of conducting a stereoconvergent cycloaddition reaction. Additional control experiments showed that alkene photoisomerization was not responsible for the stereochemical outcome observed with **5r** and **5s**. Cyclopentene, cyclohexene, and norbornene led to multi-fused cyclic products **5t**, **5u**, and **5v** in 40%, 60%, and 80% yields, respectively. The crystal structure of **5u** unambiguously confirmed these compounds as in *cis*-configuration. Moreover, 1,1-disubstituted alkene could react smoothly to give product **5w** with a quarternary carbon center in high efficiency. Substrates with either methyl or cyclohexyl substituent at

the 4-position of the phenyl ring respectively gave **5x** in 70% and **5y** in 90% yield. Decreasing the electron density on the phenyl group slowed down the cycloaddition and yields were attenuated (**5z**). Regarding ylides containing more than one nucleophilic site, the reaction afforded products as a mixture of isomers (**5aa, ab**). Furthermore, our strategy permitted the use of a diverse range of heteroarenes, enabling [4 + 2] processes with thiophenyl (**5ac, ad**), benzothiophenyl (**5ae**), and furyls (**5af**).

When styrene derivatives were used as a C$_2$ synthon in the photocyclization reaction, instead of getting the 6-membered ring sulfoxonium ylides, we exclusively isolated dihydrofuranyl sulfoxonium products (Fig. 4). After irradiating with blue LED for 3 h, **3c** reacted with 1,1-diphenylethylene to give tri-substituted dihydrofuran **7a** in 95%

**Table 1 | Optimization studies of [4 + 2] cycloaddition reaction with olefine[a]**

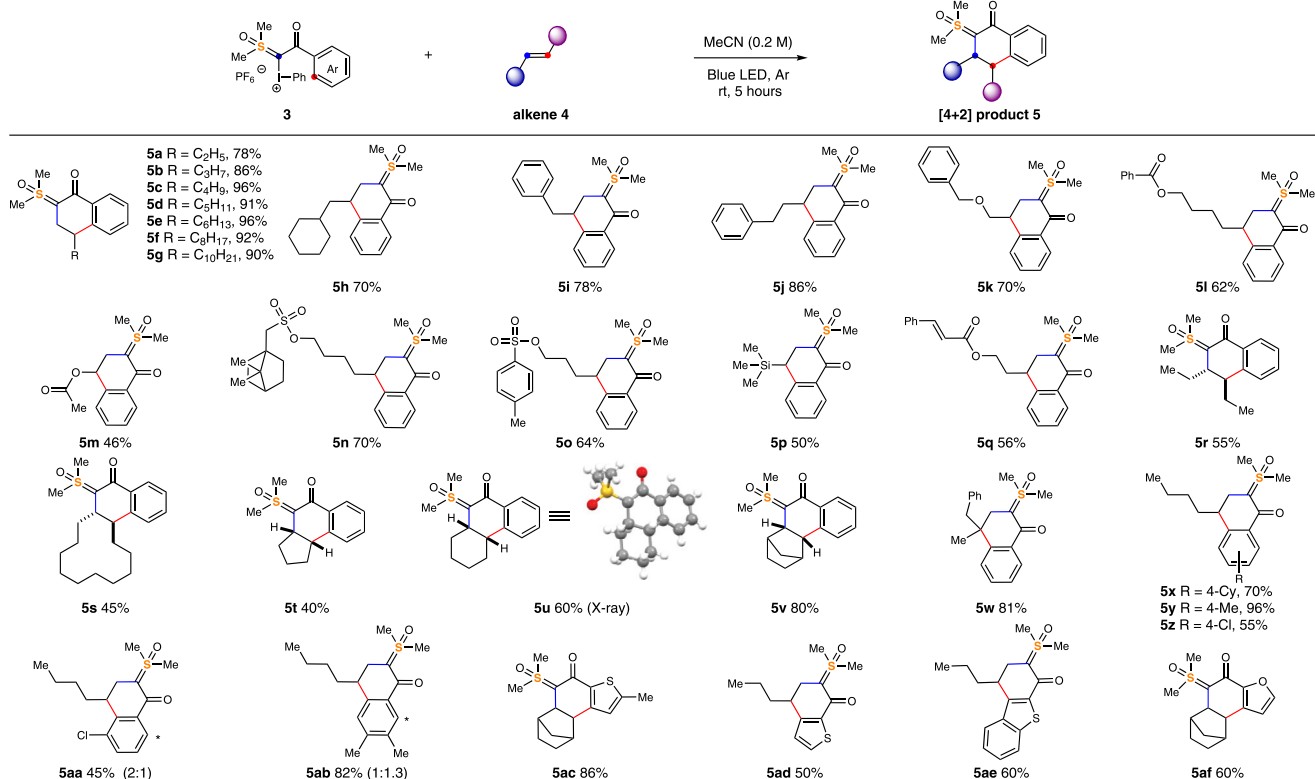

| entry | I(III)/S(IV) reagent | solvent | Yield (%)[b] |
|---|---|---|---|
| 1 | **3c** | MeCN | 95 |
| 2 | **3d** | MeCN | 55 |
| 3 | **3e** | MeCN | 50 |
| 4 | **3a** | MeCN | – |
| 5 | **3c** | DCM | 30 |
| 6 | **3c** | MeOH | – |
| 7[c] | **3c** | MeCN | 10 |
| 8[d] | **3c** | MeCN | – |
| 9[e] | **3c** | MeCN | 90 |
| 10[f] | **3c** | MeCN | 93 |
| 11[g] | **3c** | MeCN | 96 |

[a]Performed with 1-hexene (0.8 mmol, 4 equiv.), reagent **3** (0.2 mmol, 1 equiv.), solvent 1.0 mL (0.2 M).
[b]Isolated yields.
[c]Open air condition.
[d]Reaction performed in the dark.
[e]0.4 mmol of 1-hexene (2 equiv.) was used.
[f]Irradiation with white LED.
[g]5 h reaction time.

**Fig. 3 | The scope of visible-light-mediated [4 + 2] cycloaddition with aliphatic alkenes.** Reactions were performed by mixing alkene (0.8 mmol, 4 equiv.) and reagent **3** (0.2 mmol, 1 equiv.) in 1.0 mL of MeCN at room temperature, under irradiation of blue LED for 5 h, isolated yields. See Supplementary Methods for experimental details.

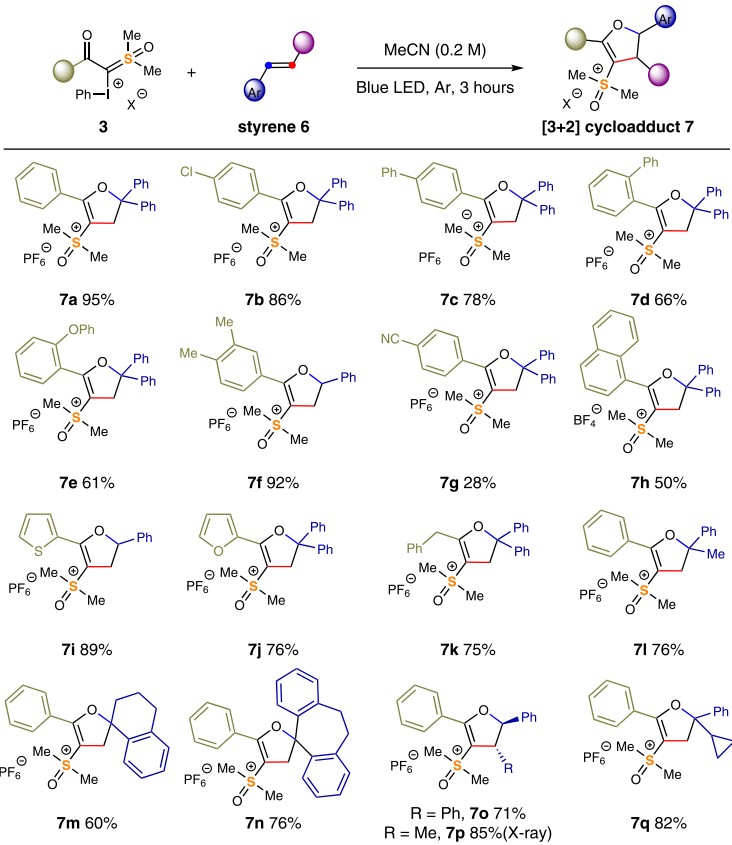

**Fig. 4 | The scope of visible-light-mediated [3+2] cycloaddition of styrenes.** Reactions were performed by mixing styrene (0.4 mmol, 2 equiv.) and reagent **3** (0.2 mmol, 1 equiv.) in 1.0 mL of MeCN at room temperature, under irradiation of blue LED for 3 h, isolated yields. See Supplementary Methods for experimental details.

yield. Other styrene derivatives, including those containing halogen (**7b**), phenyl (**7c, d**), ether (**7e**), and alkyl (**7f**) were well tolerated. It is possible that during this [3 + 2] cyclization event, the stability of the in situ generated carbocation dictated the cyclization point. When styrene was employed, the more stable benzylic cation reacted with the oxygen nucleophile (ketone), while a more reactive alkyl cation intermediate was preferentially incepted by the aromatic π-electrons in a Friedel-Crafts manner[59,60]. Ylides bearing a strong electron-withdrawing cyano-group exhibited poor reactivity, resulting in a 28% yield of **7g**. This procedure was successfully applied to a wide range of reagents **3** functionalized with 1-naphthyl (**7h**), thiophenyl (**7i**), furyl (**7j**), and benzyl (**7k**). 1,1-Disubstituted olefins, which are commercially available or easily accessible from acetophenone, were suitable substrates (**7l-n**). When 1,2-disubstituted styrenes were used, a synthetic challenging tetrasubstituted dihydrofuran scaffold was pre-pared (**7o** and **7p**) It is noteworthy that this [3 + 2] process is steor-eoconvergent and only the *trans*-isomers were obtained exclusively, which was confirmed by X-ray diffraction analysis of **7p**. Stilbene and β-methylstyrene with either olefin conformation (*E* or *Z*) delivered trans-products only (**7o** and **7p**, respectively). (1-Cyclopropylvinyl)benzene reacted smoothly with **3c** to the desired [3 + 2] product **7q**, with no ring-opening product detected. This result suggests the corresponding benzylic radical was unlikely involved in this process.

To further explore the reactivity of reagent **3** under visible light radiation, we tested reactions against other unsaturated bonds, such as nitriles, alkynes, and allenes (Fig. 5). Due to the inert reactivity of nitriles, similar cyclization reactions under blue LED did not happen. To our delight, subjecting the I, S-ylide to nitrile as solvent under the irradiation of purple LED or in the presence of Rh₂(OAc)₄ at 25 °C led to a smooth conversion to the [3 + 2] cycloaddict, isoxazolyl sulfox-onium salts (**9a-f**) in high yields. More reactive alkynes were converted

to multisubstituted furans following the [3 + 2] pathway (**10a-c**). Reactions of alkyl acetylenes afforded cyclic vinyl,ketyl-disubstituted sulfoxonium ylides via [4 + 2] (**11a-c**). When allenes were tested, the [4 + 2] cycloaddition at the terminal double bond occurred, resulting in product **13** in 72% yield. It is noteworthy that the reactions involving alkynes and allenes were carried out using acetonitrile as solvent indicating the reactivity of I⁽ᴵᴵᴵ⁾/S⁽ⱽᴵ⁾ reagents was highly selective toward carbon-carbon triple bonds. Structurally sophisticated sulfoxonium molecules could be conveniently prepared, which would be highly challenging using existing methods.

## Mechanistic investigations

Mechanistic aspects of the photocycloaddition were investigated. For the [4 + 2] reaction using aliphatic alkenes, no desired products were formed in the presence of metal catalysts (Rh and Cu) or under dark conditions. When TEMPO was added as a radical scavenger, the desired reaction was suppressed (Fig. 6a, left). In contrast, the [3 + 2] cycloaddition involving 1,1-diphenylethylene (**6a**) could be promoted by these transition metals in the absence of light, albeit in slightly decreased yields (85% using Rh and 60% using Cu). Interestingly, the [3 + 2] cycloaddition was not significantly affected by TEMPO (Fig. 6a, right). These results indicated that two distinct reaction pathways were operative when aliphatic olefins and styrenes were used, respectively. Likely, the [4 + 2] cycloaddition involved radical intermediates and the [3 + 2] reaction was carbene/carbenoid mediated. The complete che-mical reactivity was remarkable. The stereodivergent nature of both cycloadditions, when *E* and *Z*-alkenes were used, indicated a stepwise cyclization. Furthermore, we observed a substantial decrease in yield when I⁽ᴵᴵᴵ⁾/S⁽ⱽᴵ⁾ reagents contained an electron-withdrawing group on the aryl moiety (Fig. 6b), suggesting the involvement of cations. We carried out a photo-promoted [4 + 2] reaction using an allyl amide

**Fig. 5 | Cycloaddition of other unsaturated bonds.** Condition A: Reagent **3** (0.2 mmol, 1 equiv.) in 1.0 mL of nitrile solvents, under irradiation of 40 W purple LEDs for 3 h at RT; Condition B: Reagent **3** (0.2 mmol, 1 equiv.) and Rh₂(OAc)₄ (2 mol%) in 1.0 mL of nitrile solvents, RT, 10 min; Condition C: Alkynes (0.4 mmol, 2 equiv.)/allene (0.4 mmol, 2 equiv.) with reagent **3** (0.2 mmol, 1 equiv.) in 1.0 mL of MeCN at room temperature, under irradiation of blue LED for 3 h. Yields are those of isolated products. See Supplementary Methods for experimental details.

substrate, in which the amide oxygen would intercept a cation, but not a radical. As expected, the reaction produced the corresponding oxazoline product **13** cleanly (Fig. 6c). We suspected that the cation species for both reaction pathways might be formed by the heterolytic opening of a donor-acceptor cyclopropane intermediate formed by photo-promoted addition of I, S-ylide to olefin. Unfortunately, we could not isolate this key species in these intermolecular reactions. Interestingly, intramolecular reactions did deliver the desired cyclopropane products **14**, providing support for this hypothesis (Fig. 6d). Further studies are currently underway to fully elucidate the mechanistic details for the carbon-carbon forming processes and substrate-controlled reaction pathway divergence.

## Synthetic derivatizations

The sulfoxonium products were demonstrated as versatile substrates for various transformations. As shown in Fig. 7, product **5i** was reduced to cyclic ketone **15** using Raney-Nickel in refluxing isopropanol. Protonation of sulfoxonium ylide **5i** with HOTf gave the corresponding sulfoxonium salt (**16**) in 90% yield and with a 1:1.5 dr ratio. The in situ generated sulfoxonium salt was converted into sulfide **17** in high dr using a Brønsted acid. Compound **7a** was a competent electrophile in palladium-catalyzed cross-coupling reactions, leading to the formation of coupled product **18**. The sulfoxonium functionality was demethylated by heating with KI in acetone, giving sulfoxide **19**. Additionally, sulfoxonium could undergo SET processes using photoredox catalysis to generate the corresponding vinyl radical, which was functionalized to give vinyl dihydrofuran **20** in 70% yield. This oxidative radical polar crossover reaction was a rare example of using sulfoxonium as an oxidative radical precursor[61-63].

## Discussion

We designed and synthesized an I(III)/S(VI) reagent for sulfoxonium transfer reactions. These molecules revealed mixed I, S-ylide characteristics and demonstrated remarkable photocycloaddition reactivity towards a broad scope of unsaturated bonds under ambient blue LED radiation without the need for any catalyst and photosensitizer. Unusual chemical selectivity was observed, which was controlled by the type of unsaturated bonds. For aliphatic substituted double and triple bonds, a formal [4 + 2] cycloaddition occurred to give cyclic ketyl alkyl/vinyl sulfoxonium ylide products. When aryl-substituted unsaturated bonds were employed, a [3 + 2] prevailed, delivering structurally exotic cyclic vinyl sulfoxonium salts. The diverse reactivity of the sulfoxonium functionality in the products could be exploited in further chemical derivatizations.

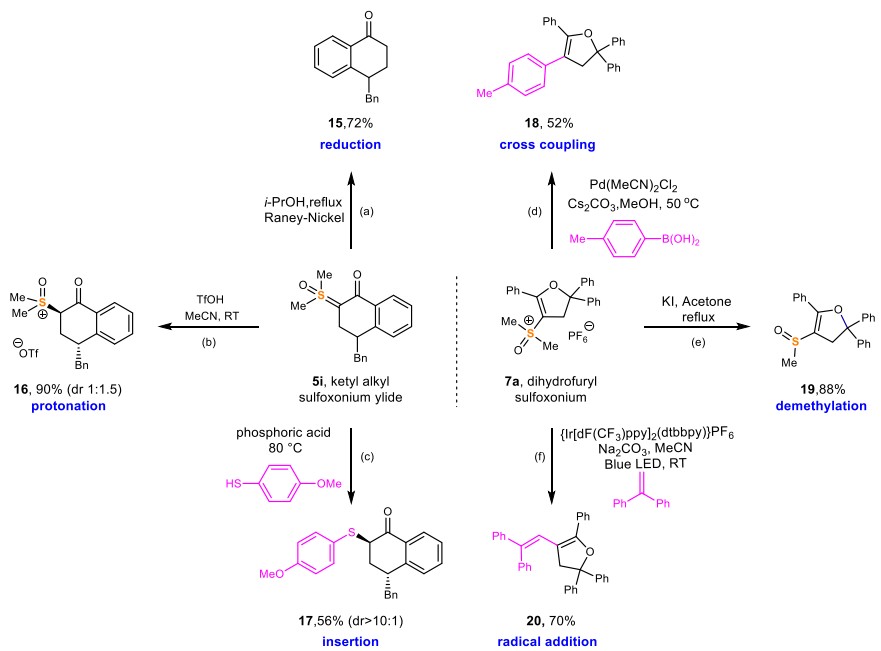

**Fig. 6 | Mechanistic study. a** Control experiment with metal catalysts and radical trapping experiment with TEMPO (2,2,6,6-tetramethylpiperidinooxy); **b** Substitutions effect; **c** Cation trap experiment; **d** Intramolecular cyclopropanation; **e** Proposed reaction mechanism.

**Fig. 7 | Synthetic transformations of sulfoxonium containing products.**
**a** Raney-Nickel, i-PrOH, reflux. **b** TfOH, MeCN, RT. **c** Phosphoric acid, 4-methoxybenzenethiol, 80 °C. **d** p-Tolylboronicacid, Pd(MeCN)₂Cl₂, Cs₂CO₃, MeOH, 50 °C.
**e** Potassium iodide, acetone, reflux. **f** {Ir[dF(CF₃)ppy]₂(dtbbpy)}PF₆, 1,1-diphenylethylene, Na₂CO₃, MeCN, Blue LED, RT. See Supplementary Methods for experimental details.

## Methods

### General procedure for photochemical cycloaddition reaction with I(III)/S(VI) reagents

In a typical experiment, to a 10 mL oven-dried tube equipped with a stirring bar was added reagent **3e** (108.8 mg, 0.2 mmol), cyclohexene **4u** (65.7 mg, 0.8 mmol), anhydrous MeCN (1.0 mL). The reaction vial was capped with a rubber septum under an argon atmosphere, and it

was fixed on a SynLED 4 × 4 photoreactor (SynLED discover™ 452 nm, 1 W, designed and manufactured by Shenzhen SynLED Tech. Ltd.) for 5 h. The solvent was removed under vacuum and the crude mixture was purified by flash column chromatography to yield the corresponding products **5u** (33.1 mg, 60%). The reactions with other kinds of unsaturated compounds were carried out similarly and the procedures are presented in Supplementary Methods.

## Data availability

Detailed experimental procedures and characterization of structurally novel compounds, mechanistic studies, and DFT calculation results are available in Supplementary Methods. For NMR spectra of structurally novel compounds, see Supplementary Figures. The crystallographic data generated in this study have been deposited at the Cambridge Crystallographic Data Centre (CCDC) under deposition numbers 2068912 (**3b**), 2068911 (**3e**), 2128359 (**5 u**), 2128864(**7p**) 2152681 (**9e**), 2144537 (**11a**). These crystallographic data can be obtained free of charge from the Cambridge Crystallographic Data Centre via http://www.ccdc.cam.ac.uk/data_request/cif. Further relevant data are available from the corresponding author upon request.

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

## Acknowledgements

This work was funded by the National Natural Science Foundation of China (21825101, Y.H.; 22101079, Z.W.), Hong Kong RGC (16300320, Y.H.), and Shenzhen Science and Technology Innovation Commission (SGDX2019081623241924, Y.H.). Z.W. thanks the National Program for Thousand Young Talents of China. We thank Dr. Herman Sung from HKUST and Dr. Xiaoli Pei from Tokyo University for the X-ray diffraction analysis.

## Author contributions

Y.H. and Z.W. conceived and directed the project. L.L., K.D., Y.X., and Z.W. performed the synthetic experiments and analyzed the experi-mental data. C.M. and S.N. performed the DFT calculations. Y.H. and Z.W. wrote the manuscript with input from all authors. All authors have read and approved the final manuscript.

## Competing interests

The authors declare no competing interests.
