## [Peer Review File · Nature Communications]

Synthesis of I(III)/S(VI) Reagents and Their Reactivity in Photochemical Cycloaddition Reactions with Unsaturated BondsREVIEWER COMMENTS

Reviewer #1 (Remarks to the Author):

In this manuscript, Profs. Huang, Wang and coworkers reported the synthesis of a novel I(III)/S(VI) reagent and had it applied to the cyclization reactions with alkenes, alkynes, nitriles and allenes as a class of sulfoxonium-transferring reagent. Strikingly, the reaction demonstrates excellent chemical selectivity depending on the type of the unsaturated bonds. The work is overall well designed and implemented, and interesting results have been achieved.

S-Containing hypervalent iodine reagents are quite limited. This work presented an interesting reagent containing both hypervalent iodine and hypervalent sulfur moieties. However, as the authors have stated in the text, the closely related sulfonium-iodonium ylides have already been established by Podrugina and co-workers (ref 54). Furthermore, although the authors have carried out preliminary investigations, the current reaction mechanism is less convincing to account for the 'role' of the hypervalent iodine moiety, the necessity of photo irradiation as well as the selectivity of the reaction.

With regards to the above two points, the reviewer does not think the work is qualified enough to be published in Nature Communications.

Reviewer #2 (Remarks to the Author):

The main aspect of the method reported in the submitted manuscript is that it allows access to some previously unavailable sulfoxonium ylides or their salts.

Previously, sulfonium-iodonium ylides had been prepared but these were unstable and had to be used in-situ in a single reaction, namely with ammonium thiocyanate (ref 54). The sulfoxonium-iodonium ylides described in this manuscript are more stable and were prepared on multi-gram scale. In the presence of metal catalysts or when irradiated with violet or blue LEDs, formal cycloaddition reactions with unsaturated moieties (alkenes, alkynes, allenes, and nitriles) to give access to sulfoxonium ylides (compounds 5 in Scheme 3 and 11/13 in Scheme 5) or their salts (compounds 7 in Scheme 4 and 9 & 10 in Scheme 5). Another related precedent is diazo carboxylate - iodonium ylides reported by

Suero (Nature 2018, 554, 86) where a ethyl diazo acetate fragment replaces the alpha-keto sulfoxonium fragment in compounds 3 of the submitted manuscript.

Examples 5a–5g are basically the same compounds except that the length of the linear alkyl chain was systematically increased. These examples show an unexpectedly large variation in yield of isolated product (71–96 %), which casts a small doubt about the reproducibility of the mass transfer in these reactions. Moreover, several examples in Scheme 3 show moderate to low yield: what is the rest of the mass balance in these reaction? The authors claim that a single regioisomer was obtained, but is it also the case in the crude material before purification? Even if the regioselectivity is good (e.g. > 10:1), it would be better to include that information. Moreover, how was the structure of the major regioisomer ascertained: if HSQC and HMBC have been done to that effect, they should be included in the SI, and if not, this NMR data should be acquired to prove the structure of the products. The X-ray of 5u is not relevant for regioselectivity as the olefin is symmetrical.

In scheme 4, the authors mentioned that the 3+2 process is stereoconvergent. However, in the SI, the geometry of the stilbene precursor to 7o is not specified and a E olefin was used to make 7p. Thus, the authors cannot conclude about the stereoconvergence of the process on that basis, except if the same diastereomer of 7o and 7p could be obtained from Z olefins. Moreover, how was the relative configuration of 7o and 7p established? No evidence is provided in the SI.

Scheme 5 is misleading, as the general equation suggest that all compounds in that scheme are obtained under blue light irradiation without metal catalyst. However, compound 9a was obtained under violet light, and compounds 9b–9f were obtained in the presence of Rh₂(OAc)₄ without blue light irradiation. Please use several schemes or equations to make this less confusing: forcing the reader to look at the footnote of the scheme for such fundamental (as opposed to minor) differences is not helpful. Why was Rh(II) needed for 9b–9f? Did violet light irradiation in the absence of metal catalysts give any of 9b–9f? The SI specifies a 95% of 9a in the presence of the Rh(II) catalyst, which should be stated in the main text. How did the reaction of 3 and 4 or 6 perform under violet light irradiation?

As for other examples, regioselectivity should be evidenced for the structure of regioisomers 11a–c (i.e. prove it's not the other regioisomer). The structure of 10c (cyclohexyl) is not the same as in the SI; again provide evidence of that 10a–c is the regioisomer depicted. Ditto for the product formed by reaction with an allene: the geometry

of the olefin must be verified with spectroscopic evidence.

Scheme 7 lacks the details (reagents & conditions) of each transformation. Please include them (again, forcing the reader to look at the SI for that information is not helpful).

With respect to the mechanism, there is no sufficient evidence to conclude about the cationic mechanism alluded to in Scheme 6d. It could very well be that diradical intermediates are involved (see ref 58). Alternatively compounds 5 could be obtained in a radical mechanism similar to that described for related compounds by Suero et al in Nature 2018, 554, 86. Of course, there is no metal photocatalyst in the present study, but it is not excluded that sulfonium-iodonium ylides 3 can absorb blue light when in solution, possible as part of an electron-donor complex. It is also not excluded that a radical is further oxidised to a carbocation under the reaction conditions. A more substantial insight in the mechanism is needed for publication in this journal

Hence, the authors should

- 1) measure UV/vis spectrum of compounds 3 in MeCN and superimpose these with the emission spectrum of their LEDs.
- 2) examine the reactivity of 1,6-diene in the reaction with 3t or 3u; the 4+2 is not possible with these, and if a radical intermediate is generated from 3t or 3u, it should give a cascade reaction in a reaction with the 1,6-diene to give likely a 5-membered ring compound.
- 3) examine the reactivity of (1-cyclopropylvinyl)benzene in the reaction with 3c or 3e; if a radical intermediate is generated from 3c or 3e that reacts with that olefin, a ring-opening should occur (radical clock experiment).

In the SI, the description of the NMR data contains many errors.

- 1) a very frequent error made by the authors is to describe the two singlets of the (O=)SMe₂ fragment of compounds 5 as a doublet with variable J values (some impossibly large (> 20 Hz)) that integrates for 6H. The two methyl groups are not equivalent in compounds 5 and each gives a singlet for 3H. Please correct all instances.
- 2) AB system in compounds 5 (two AB systems in 5w) should be described in more details (i.e. not multiplet for 2 H, but 2 dd for 1 H each).
- 3) the triflate in 3d give a visible quartet in ¹³C due to the large C-F coupling, but this is not described: instead the authors list mistakenly 122.51 and 119.33 ppm as peaks for two different nuclei.

In addition, the authors should give more details about the irradiation set-up. The SI states 1 W, 452 nm for blue LED but nothing about the actual equipment and distance of the LED to the vial. How was the temperature controlled? The SI states a different wave length than what is mentioned in the main text (i.e. 452 vs 465 nm).

Other minor corrections

Drawing compounds 3 with a I-P or I-B bond should be avoided. There is no such bond with the hypervalent counteranions PF₆ or BF₄. The I...F contact in the xray structures is of course possible, but it's not a I-P or I-B bond.

As compounds 3 are the key novelty of the study, it would be good to move the scope of compounds 3 from the SI to the main manuscript.

Overall, this work can lead to important discovery in the field of sulfoxonium ylides chemistry and the wider field of organic synthesis, but major corrections are necessary before considering publication in this journal.

Reviewer #3 (Remarks to the Author):

This review considers only the X-ray diffraction analyses and associated interpretation.

The X-ray structure refinements have been carried out well and do not raise any concerns. Some more detailed comments, including suggestions for minor revision, are provided in the accompanying PDF file.

This review considers only the X-ray diffraction analyses and associated interpretation. The X-ray structure refinements have been carried out well and do not raise any concerns.

Some minor changes should be made to the refinement of **9e**. These do not have any significant consequence on the results, but they would give a fairer interpretation of the data.

- The DANG restraints applied to the BF₄ anion are specified as 2.4 Å. Comparison to other structures (including **3e** in this paper) suggests that this distance should be closer to 2.25 Å. The best strategy would probably be to link this distance to a refined free variable. If that is done, the refined value comes out to be 2.26 Å.
- There is a long list of OMIT instructions, which remove 39 reflections from the refinement. If I repeat the refinement with the DANG restraints as described above and optimise the weighting

scheme, the largest error/su for any reflection is ca 4.8. So there are no significant outliers and no reason to omit these 39 measured reflections.

There are some points to be made regarding the interpretation of the X-ray results:

- (1) Bond distances and angles come with standard uncertainties. The caption for Scheme 2, for example, should include these uncertainties. Individual values should be reproduced exactly as they appear in the CIF.
- (2) The paragraph following Scheme 2 contains several observations based on atom-atom distances in the X-ray structures of **3b** and **3e**. Some of these are not supported by the X-ray data and I think that some of the conclusions are not fully justified.
 - *“A striking feature of **3e** is the short S1...O1 distance ... which indicates an intramolecular interaction between C=O and S”.* Although the distance is short compared to the sum of the VDW radii, I do not agree that it necessarily indicates some noteworthy interaction. Much closer contacts exist between the C=O group and the H atoms of the CH₃ groups. Looking at a space-filling representation shows that the CH₃ groups effectively shield the S atom from any interaction with the O atom. What is the nature of the proposed interaction between O and S? This needs further justification.
 - *“Partial double bond connectivity between C1 and C2 and a small elongation of the C=O bond were noted.”* The expected bond distances for C-C and C=C are fairly well established, and it is clear that the C1-C2 distance is shorter than would be expected for a C-C single bond (ca 1.54 Å). So the first comment is justifiable. But what is the “small elongation of the C=O bond” compared to? Looking at the C=O distances in all of the structures in this paper, the C=O bond in **3e** is significantly **shorter** than in **5u** or **11a**. Possibly the suggestion that is being made is that the C=O bond in **3e** is longer than in **3b**? This is not supported by the X-ray data. To compare distances in X-ray crystal structures, the observed difference must be compared to the uncertainty on the difference [which is calculated as the square root of (variance(1) + variance(2))]. Comparing the C=O bond distances for **3b** vs **3e**:

3e	3b	Difference	ESD of difference	3 x ESD	Difference > 3 x ESD
1.248(5)	1.239(3)	0.009	$\sqrt{(0.005^2+0.003^2)} = 0.0058$	0.0175	NO
1.248(5)	1.230(4)	0.018	$\sqrt{(0.005^2+0.004^2)} = 0.0064$	0.0192	NO

So it's not clear what "*small elongation of the C=O bond*" actually means. Please clarify this.

- Similarly, the C1-S1 and C1-I1 bond lengths are being presented as having "*partial double bond character*". What's the basis for this? A search of the Cambridge Structural Database for C-I single bonds finds about 10,000 compounds, with a mean C-I distance of 2.10(4) Å. So the value of ca 2.06 Å seen in **3b** and **3e** is within one standard deviation of the mean C-I distance seen for all C-I single bonds in the CSD. How does this suggest partial double bond character? Likewise, the mean value for the first 10,000 C-S single bonds identified in the CSD is 1.77(5) Å. Again the values seen in the structures in this current paper are mostly around one standard deviation from this mean value. Please clarify how the distances seen in the X-ray crystal structures yield the conclusion that there is "*partial double bond character*".
- Finally, a similar question arises for the conclusion that "*the X-ray data suggest a mixed iodonium-sulfonium ylide character.*" Please just clarify this in light of the comments above.

I would suggest that it would be useful to support the conclusions on the "*mixed iodonium-sulfonium ylide character*" with a calculation of the electronic structure of the molecule. At least for the small molecule **3e**, this looks feasible, and I think it would add a great deal to the interpretation of the X-ray structures. I realise that the paper is mainly focussed on the synthetic chemistry of these molecules, but the authors do seek to comment on the nature of **3b** and **3e** by analysis of the X-ray structures, and a calculation would add to the information that can be deduced from the structures alone.

Response to reviewers:

Many thanks for the valuable revision of our manuscript entitled "Synthesis of a Class of I(III)/S(VI) Reagents and Their Reactivity in Photochemical Cycloaddition Reactions with Unsaturated Bonds". We have addressed all the points and concerns and a point-by-point response is included herein. The major changes in the revised manuscript are:

- 1) Scope of compounds **3** has been moved from SI to Scheme 2;
- 2) Scheme 5 has been redrawn to reflect the difference in reaction conditions;
- 3) Additional mechanistic experiments have been added into scheme 6, with discussions made in the corresponding part of the manuscript;
- 4) We conducted theoretical calculations to strengthen the claims related to the mixed iodonium-sulfoxonium ylide character of I(III)/S(VI) reagents;
- 5) Some spectroscopic and X-ray evidence has been provided to support claimed regioselectivity and stereochemistry of the reaction;
- 6) Some mistakes relating to characterization data and inconsistencies have been corrected.

The concerns and comments raised by reviewers #1 and #2 have been highly important (and motivational) to further expand the detailed mechanistic studies of the photochemical cyclization reactions (Scheme 6). The suggestions and comments raised by reviewer #3 lead to improved structure interpretation of I(III)/S(VI) reagents. The reviewers' comments have significantly improved this paper.

Point-to-point responses to reviewers' comments are listed on following pages (in blue).

****New products have been fully characterized and reported in the revised Supplementary Information****

Reviewer #1 (Remarks to the Author):

In this manuscript, Profs. Huang, Wang and coworkers reported the synthesis of a novel I(III)/S(VI) reagent and had it applied to the cyclization reactions with alkenes, alkynes, nitriles and allenes as a class of sulfoxonium-transferring reagent. Strikingly, the reaction demonstrates excellent chemical selectivity depending on the type of the unsaturated bonds. The work is overall well designed and implemented, and interesting results have been achieved.

Our responses: we thank reviewer 1 for his/her kind words on the overall novelty of this work.

S-Containing hypervalent iodine reagents are quite limited. This work presented an interesting reagent containing both hypervalent iodine and hypervalent sulfur moieties. However, as the authors have stated in the text, the closely related sulfonium-iodonium ylides have already been established by Podrugina and co-workers (ref 54). Furthermore, although the authors have carried out preliminary investigations, the current reaction mechanism is less convincing to account for the 'role' of the hypervalent iodine moiety, the necessity of photo irradiation as well as the selectivity of the reaction.

With regards to the above two points, the reviewer does not think the work is qualified enough to be published in Nature Communications.

Our responses: yes, there is one report on sulfonium-iodonium ylides by Podrugina and co-workers (ref 54). However, these mixed ylides reagents were highly unstable and were unable to be isolated. Little is known about the reactivity of I,S-ylides. Podrugina's paper mostly described reactivity studies using the more stable As,I-ylides.

The structure of this compound was verified by low-temperature NMR spectroscopy, since this mixed sulfonium–iodonium ylide **6a** is unstable at temperatures above $-10\text{ }^{\circ}\text{C}$. The composition of the ylide **6a** was verified by HRMS spectroscopy. However, oxidation of sulfonium ylides **5b** and **5c** resulted in formation of less stable compounds, which we were not able to characterize. Ref. 54

In sharp contrast, our sulfoxonium-iodonium ylide reagents are much more stable and could be easily isolated as solids. Our subsequent studies showed the unique reactivity of I,S-ylides against olefins under photoirradiation. We have carried out further mechanistic studies. Both the hypervalent iodide moiety and photo-irradiation were required.

We hypothesized a photo-promoted cyclopropane intermediate was responsible for both [4+2] (for alkyl olefins) and [3+2] (for styrenes) products. However, the corresponding cyclopropane intermediate was highly unstable and could not be detected. Removing the phenyl ring next to the carbonyl also failed to give a stable cyclopropane product. Interestingly, intramolecular reactions did deliver the desired cyclopropane products, supporting cyclopropanation being the first step under photo conditions. Based on these results, we propose the following mechanism. For intermolecular reactions, the unstable three-membered intermediate might have been formed first, which is a highly reactive donor-acceptor (DA) cyclopropane and is prone to selective bond-opening to give intermediates A and B (for alkyl olefin and styrene substrates, respectively). The resulting cation A is unstabilized (quite reactive) and undergoes Friedel-Crafts alkylation to give the [4+2] product. Cation B is stabilized by the neighboring ph group and reacts preferably with the enolate form to give the [3+2] product. These experimental data and discussions have been added to the revised manuscript and SI.

Reviewer #2 (Remarks to the Author):

The main aspect of the method reported in the submitted manuscript is that it allows access to some previously unavailable sulfoxonium ylides or their salts. Previously, sulfonium-iodonium ylides had been prepared but these were unstable and had to be used in-situ in a single reaction, namely with ammonium thiocyanate (ref 54). The sulfoxonium-iodonium ylides described in this manuscript are more stable and were prepared on multi-gram scale. In the presence of metal catalysts or when irradiated with violet or blue LEDs, formal cycloaddition reactions with unsaturated moieties (alkenes, alkynes, allenes, and nitriles) to give access to sulfoxonium ylides (compounds 5 in Scheme 3 and 11/13 in Scheme 5) or their salts (compounds 7 in Scheme 4 and 9 & 10 in Scheme 5). Another related precedent is diazo carboxylate - iodonium ylides reported by Suero (Nature 2018, 554, 86) where an ethyl diazo acetate fragment replaces the alpha-keto sulfoxonium fragment in compounds 3 of the submitted manuscript.

Our responses: this reviewer gave a very accurate summary of this work. We thank him/her for his/her valuable time to carefully read this manuscript.

Examples 5a–5g are basically the same compounds except that the length of the linear alkyl chain was systematically increased. These examples show an unexpectedly large variation in yield of isolated product (71–96 %), which casts a small doubt about the reproducibility of the mass transfer in these reactions. Moreover, several examples in Scheme 3 show moderate to low yield: what is the rest of the mass balance in these reactions?

Our responses: yields reported in scheme 3 were the average of two runs. The moderate yield of **5a** may be due to the low solubility of 1-butene. We were puzzled by the variation of yield for compounds **5b-5g** they are very similar, only different from the length of the alkyl chain. After carefully re-examining the reaction conditions, we found the discrepancy in yield arose from residual methanol in the precursors (**3b-3g**), which were precipitated from methanol. The methanol contamination affected the yields of 5b-5g. We repeated the preparation of **5d, 5f, 5g**, and yields improved significantly. These data have been updated in the revised version.

In Scheme 3, the major side product was identified as the corresponding \$\alpha\$ -ketosulfoxonium ylide. It was a common decomposition pathway of I,S-ylides. The exact reason for this side reaction was not clear.

The authors claim that a single regioisomer was obtained, but is it also the

case in the crude material before purification? Even if the regioselectivity is good (e.g. > 10:1), it would be better to include that information.

Our Response: For all the substrates listed in scheme 3, we did not observe the other regioisomer from the crude reaction mixture using ^1H NMR spectroscopy. We believe this cyclization reaction is highly regio-selective. This information will be updated in the revised SI.

Moreover, how was the structure of the major regioisomer ascertained: if HSQC and HMBC have been done to that effect, they should be included in the SI, and if not, this NMR data should be acquired to prove the structure of the products. The X-ray of 5u is not relevant for regioselectivity as the olefin is symmetrical.

Our response: To confirm the structure of the major regioisomer, we did a reduction of **5i** to afford known compound **14**. The NMR data of the reduced product is identical to the reported one (ref: *Org. Lett.* 2011, 13, 3266–3269, compound **6a**, page 31 on Supplementary Information), in this way we confirmed the regioselectivity as shown.

We have added this reference to our revised Supplementary Information.

[3] b) Quiclet-Sire, B. et. al. Oxime Derivatives as α -Electrophiles. From α -Tetralone Oximes to Tetracyclic Frameworks. *Org. Lett.* **2011**, 13, 3266–3269.

In scheme 4, the authors mentioned that the 3+2 process is stereoconvergent. However, in the SI, the geometry of the stilbene precursor to 7o is not specified and a E olefin was used to make 7p. Thus, the authors cannot conclude about the stereoconvergence of the process on that basis, except if the same diastereomer of 7o and 7p could be obtained from Z olefins. Moreover, how was the relative configuration of 7o and 7p established? No evidence is provided in the SI.

Our responses: for internal styrenes, the reaction produced the trans-[4+2] product only. We used pure E and Z olefins to perform the cycloaddition separately, and only the trans-products were detected.

The relative configuration of **7p** was confirmed by X-ray diffraction analysis (CCDC 2128864). The relative stereochemistry of **7o** was assigned based on the value of the similar coupling constant between two vicinal C-H in the dihydrofuran ring.

Scheme 5 is misleading, as the general equation suggest that all compounds in that scheme are obtained under blue light irradiation without metal catalyst. However, compound **9a** was obtained under violet light, and compounds **9b–9f** were obtained in the presence of $\text{Rh}_2(\text{OAc})_4$ without blue light irradiation. Please use several schemes or equations to make this less confusing: forcing the reader to look at the footnote of the scheme for such fundamental (as opposed to minor) differences is not helpful. Why was Rh(II) needed for **9b–9f**? Did violet light irradiation in the absence of metal catalysts give any of **9b–9f**? The SI specifies a 95% of **9a** in the presence of the Rh(II) catalyst, which should be stated in the main text. How did the reaction of **3** and **4** or **6** perform under violet light irradiation?

Our responses: Scheme 5 has been redrawn to reflect the difference in reaction conditions. Reactions of I,S-ylides with nitriles were very slow under blue LED irradiation and that is why we could use MeCN as solvent for Schemes 3 and 4. We found 40W purple LEDs ($\lambda_{\text{max}}390$ nm) were able to promote the reaction with MeCN. However, the yield for higher order nitriles was low to moderate. Alternatively, these substrates performed exceedingly well in the presence of $\text{Rh}_2(\text{OAc})_4$.

9a, 67%

9b, trace

9c, 35%

9d, 55%

9e, 62%

9f, trace

As for other examples, regioselectivity should be evidenced for the structure of regioisomers 11a–c (i.e. prove it's not the other regioisomer). The structure of 10c (cyclohexyl) is not the same as in the SI; again provide evidence of that 10a–c is the regioisomer depicted. Ditto for the product formed by reaction with an allene: the geometry of the olefin must be verified with spectroscopic evidence.

Our responses: The regiochemistry of **11a** was confirmed by X-ray analysis (CCDC 2144537). The stereochemistry of **11b** and **11c** was assigned accordingly by comparing the chemical shift of alpha proton of the sulfoxonium ylide.

The regiochemistry of product **10a** was assigned by NOE experiments, which revealed the proximity of the following two Me groups. This information has been added to the revised SI. The regiochemistry of products **10b** and **10c** were assigned accordingly.

After careful examination of product 13 using allene, we found the product was a mixture of two isomers (6.3:1). The major isomer was assigned to be Z according to J values of Ha and Hb. In the major isomer, the coupling constant of Ha and Hb is 1.8 Hz) and as a result, the two Hb showed as a doublet. In the minor isomer, the singlet peak of Hb indicates there are no protons nearby. We have corrected this data in our revised manuscript.

major: Z isomer

minor: E isomer

5.77 (dt, $J = 8.9, 1.8$ Hz, $1H^a$)

5.47 (d, $J = 10.2$ Hz Hz, $0.16 \cdot 1H^a$)

3.43 (d, $J = 1.8$ Hz, $2H^b$)

3.20 (s, $0.32 \cdot 2H^b$)

$J_{H^a-H^c} = 8.9$ Hz

$J_{H^a-H^c} = 10.2$ Hz

$J_{H^b-H^a} = 1.8$ Hz

$J_{H^b-H^a}$ not detected

Scheme 7 lacks the details (reagents & conditions) of each transformation. Please include them (again, forcing the reader to look at the SI for that information is not helpful).

Our response: reaction details have been added in Scheme 7.

With respect to the mechanism, there is no sufficient evidence to conclude about the cationic mechanism alluded to in Scheme 6d. It could very well be that diradical intermediates are involved (see ref 58). Alternatively compounds 5 could be obtained in a radical mechanism similar to that described for related compounds by Suero et al in Nature 2018, 554, 86. Of course, there is no metal photocatalyst in the present study, but it is not excluded that sulfonium-iodonium ylides 3 can absorb blue light when in solution, possible as part of an electron-donor complex. It is also not excluded that a radical is further oxidised to a carbocation under the reaction conditions. A more substantial insight in the mechanism is needed for publication in this journal

Our responses: to support the cationic ring closure proposed in Scheme 6d, we carried out photo-induced [4+2] reaction using an allylic amide substrate, in which the amide group would intercept the cation, but not a radical. As expected, the reaction produced the corresponding oxazoline product cleanly.

Regarding the addition of I,S-ylide to olefin, the most likely intermediate would be the corresponding cyclopropane. However, the cyclopropane intermediate was highly unstable and could not be detected. Removing the phenyl ring next to the carbonyl also failed to give a stable cyclopropane product. Interestingly, intramolecular reactions did deliver the desired cyclopropane products, supporting cyclopropanation being the first step under photo conditions. Based on these results, we propose the following mechanism. For intermolecular reactions, the unstable three-membered intermediate might have been formed first, which is a highly reactive donor-acceptor (DA) cyclopropane and is prone to selective close-shell bond-opening to give intermediates A and B (for alkyl olefin and styrene substrates, respectively). The resulting cation A is unstabilized (quite reactive) and undergoes Friedel-Crafts alkylation to give the [4+2] product. Cation B is stabilized by the neighboring ph group and reacts preferably with the enolate form to give the [3+2] product. These experimental data and discussions have been added to the revised manuscript and SI.

Hence, the authors should

1) measure UV/vis spectrum of compounds **3** in MeCN and superimpose these with the emission spectrum of their LEDs.

2) examine the reactivity of 1,6-diene in the reaction with **3t** or **3u**; the 4+2 is not possible with these, and if a radical intermediate is generated from **3t** or **3u**, it should give a cascade reaction in a reaction with the 1,6-diene to give likely a 5-membered ring compound.

3) examine the reactivity of (1-cyclopropylvinyl)benzene in the reaction with **3c** or **3e**; if a radical intermediate is generated from **3c** or **3e** that reacts with that olefin, a ring-opening should occur (radical clock experiment).

Our responses: We measured UV/vis spectra of compounds **3c** and **3t** in MeCN. When overlaid with the emission spectrum of the RGB LEDs (*Opt. Express* **2018**, *26*, 18279–18291.), it was reasonable to assume that the excitation of **3c** and **3t** can occur in the blue region. These results has been added to the mechanistic experiments part in Supplementary Information.

We examined the reactivity of several diene substrates with **3u**. Unfortunately, no cascade products were observed. Most sulfoxonium-iodonium reagents decomposed and we can detect a significant amount of corresponding α -ketosulfoxonium ylides **2u**. These results also disfavored the radical mechanism.

The following radical experiments were performed.

(1-cyclopropylvinyl)benzene reacted smoothly with **3c** to give the desired [3+2] product **7q**, with no ring-opening product detected. This result suggests the corresponding benzylic radical was unlikely involved. Two other cyclopropanes were prepared and tested. The reactions were quite messy and no identifiable products were isolated.

In the SI, the description of the NMR data contains many errors.

1) a very frequent error made by the authors is to describe the two singlets of the (O=)SMe₂ fragment of compounds 5 as a doublet with variable J values (some impossibly large (> 20 Hz)) that integrates for 6H. The two methyl groups are not equivalent in compounds 5 and each gives a singlet for 3H. Please correct all instances.

Our responses: we thank this reviewer for his/her careful proofreading. He/she is right. We have corrected the characterization data for all products.

2) AB system in compounds 5 (two AB systems in 5w) should be described in more details (i.e. not multiplet for 2 H, but 2 dd for 1 H each).

Our responses: we have double-checked these data and have made corrections to the AB patterns.

3) the triflate in 3d give a visible quartet in ¹³C due to the large C-F coupling, but this is not described: instead the authors list mistakenly 122.51 and 119.33 ppm as peaks for two different nuclei.

Our response: we have corrected this error, as well as others related to C-F coupling.

In addition, the authors should give more details about the irradiation set-up. The SI states 1 W, 452 nm for blue LED but nothing about the actual equipment and distance of the LED to the vial. How was the temperature controlled? The SI states a different wave length than what is mentioned in the main text (i.e. 452 vs 465 nm).

Our responses: photos of the reaction setup were taken and included in the revised SI. The temperature was controlled by an integrated fan under the LED panel. The wavelength of blue LEDs was double-checked, and it was 452nm. Inconsistencies have been corrected.

Other minor corrections

Drawing compounds **3** with a I-P or I-B bond should be avoided. There is no such bond with the hypervalent counteranions PF₆ or BF₄. The I...F contact in the xray structures is of course possible, but it's not a I-P or I-B bond. As compounds **3** are the key novelty of the study, it would be good to move the scope of compounds **3** from the SI to the main manuscript.

Our response: the reviewer was correct. There were no interactions between P and I. The structures have been redrawn to properly reflect it. The scope of **3** has been moved from SI to the main text.

Overall, this work can lead to important discovery in the field of sulfoxonium ylides chemistry and the wider field of organic synthesis, but major corrections are necessary before considering publication in this journal.

Our responses: we would like to thank this reviewer for his/her kind words on the importance of this work.

Reviewer #3 (Remarks to the Author):

This review considers only the X-ray diffraction analyses and associated interpretation.

The X-ray structure refinements have been carried out well and do not raise any concerns.

Our responses: we thank this reviewer for approving the overall quality of X-ray data, pending minor revisions.

Some minor changes should be made to the refinement of **9e**. These do not have any significant consequence on the results, but they would give a fairer interpretation of the data.

- The DANG restraints applied to the BF₄ anion are specified as 2.4 Å. Comparison to other structures (including **3e** in this paper) suggests that this distance should be closer to 2.25 Å. The best strategy would probably be to link this distance to a refined free variable. If that is done, the refined value comes out to be 2.26 Å. □
- There is a long list of OMIT instructions, which remove 39 reflections from the refinement. If I repeat the refinement with the DANG restraints as

described above and optimise the weighting scheme, the largest error/su for any reflection is ca 4.8. So there are no significant outliers and no reason to omit these 39 measured reflections.

Our responses: we have repeated the refinement work of **9e** following these suggestions and uploaded the revised CIF file to the Cambridge Crystallographic Data Centre.

There are some points to be made regarding the interpretation of the X-ray results:

(1) Bond distances and angles come with standard uncertainties. The caption for Scheme 2, for example, should include these uncertainties. Individual values should be reproduced exactly as they appear in the CIF.

Our responses: standard uncertainty data have been added in the caption of scheme 2 according to the reviewer's suggestion.

(2) The paragraph following Scheme 2 contains several observations based on atom-atom distances in the X-ray structures of **3b** and **3e**. Some of these are not supported by the X-ray data and I think that some of the conclusions are not fully justified.

- "A striking feature of **3e** is the short S1...O1 distance ... which indicates an intramolecular interaction between C=O and S". Although the distance is short compared to the sum of the VDW radii, I do not agree that it necessarily indicates some noteworthy interaction. Much closer contacts exist between the C=O group and the H atoms of the CH₃ groups. Looking at a space-filling representation shows that the CH₃ groups effectively shield the S atom from any interaction with the O atom. What is the nature of the proposed interaction between O and S? This needs further justification.

Our responses: the statement of O-S interactions was made according to Moriarty's description of these mixed ylides (Ref. 49: *J. Am. Chem. Soc.* **1984**, *106*, 6082–6084. The reviewer was right. There were few interactions between S and C=O, we have removed the discussions of interactions between O and S.

- "Partial double bond connectivity between C1 and C2 and a small elongation of the C=O bond were noted." The expected bond distances for

C-C and C=C are fairly well established, and it is clear that the C1-C2 distance is shorter than would be expected for a C-C single bond (ca 1.54 Å). So the first comment is justifiable. But what is the “small elongation of the C=O bond” compared to? Looking at the C=O distances in all of the structures in this paper, the C=O bond in **3e** is significantly shorter than in **5u** or **11a**. Possibly the suggestion that is being made is that the C=O bond in **3e** is longer than in **3b**? This is not supported by the X-ray data. To compare distances in X-ray crystal structures, the observed difference must be compared to the uncertainty on the difference [which is calculated as the square root of (variance(1) + variance(2))]. Comparing the C=O bond distances for **3b** vs **3e**:

3e	3b	Difference	ESD of difference	3 x ESD	Difference > 3 x ESD
1.248(5)	1.239(3)	0.009	$\sqrt{(0.005^2+0.003^2)} = 0.0058$	0.0175	NO
1.248(5)	1.230(4)	0.018	$\sqrt{(0.005^2+0.004^2)} = 0.0064$	0.0192	NO

So it's not clear what “*small elongation of the C=O bond*” actually means. Please clarify this.

Our responses: the discussions of “Partial double bond connectivity between C1 and C2 and a small elongation of the C=O bond were noted.” referred to the resonance structure of the I,S-ylide (shown below). According to these resonance structures, the C1-C2 single bond should be shorter than common Csp2-Csp2, and C=O should be longer than those without such conjugating resonance. As a comparison, we examined the bond lengths of **3b** and 4-aminoacetophenone, which also exhibits weak resonance. The C=O of **3b** is longer than that of 4-aminoacetophenone (1.230 vs 1.215) and C-C bond of **3b** is shorter (1.431 vs 1.468). As a result, these I,S-ylides likely exist in multiple resonance structures.

- Similarly, the C1-S1 and C1-I1 bond lengths are being presented as having “partial double bond character”. What’s the basis for this? A search of the Cambridge Structural Database for C-I single bonds finds about 10,000 compounds, with a mean C-I distance of 2.10(4) Å. So the value of ca 2.06 Å seen in **3b** and **3e** is within one standard deviation of the mean C-I distance seen for all C-I single bonds in the CSD. How does this suggest partial double bond character? Likewise, the mean value for the first 10,000 C-S single bonds identified in the CSD is 1.77(5) Å. Again the values seen in the structures in this current paper are mostly around one standard deviation from this mean value. Please clarify how the distances seen in the X-ray crystal structures yield the conclusion that there is “*partial double bond character*”.

Our responses: the reviewer was correct. From the X-ray data, C-I is a single bond. In organic chemistry community, this type of ylide-like C-I bonds are often drawn as C=I (incorrectly) to demonstrate the iodine ylide reactivity of these molecules. We have removed the statement of “partial double bond character” for the C-I bond. On the other hand, the C-S bond did show double bond character. In the X-ray structure, the bond length of the C-S (of **3e**) is 1.716, significantly shorter than the common 1.76-1.77 for most C-S single bonds. Below is the comparison of a reported structure of a sulfoxonium ylide with that of compound **3e**. The bond length of C-S in **3e** is actually slightly shorter than the C=S bond in the reported sulfoxonium ylide (1.716 vs 1.730). In addition, the C-Me bond in **3e** shows a true C-S single bond character, with a bond length of 1.764. Therefore, the C-S bond in our I,S-ylide is exhibiting a double bond character, as sulfoxonium ylide.

CCDC 1899265

C=S (ylide) = 1.730

C=S (ylide) = 1.716

Me-S (single bond) = 1.764

- Finally, a similar question arises for the conclusion that "the X-ray data suggest a mixed iodonium-sulfonium ylide character." Please just clarify this in light of the comments above.

I would suggest that it would be useful to support the conclusions on the "mixed iodonium-sulfonium ylide character" with a calculation of the electronic structure of the molecule. At least for the small molecule **3e**, this looks feasible, and I think it would add a great deal to the interpretation of the X-ray structures. I realise that the paper is mainly focussed on the synthetic chemistry of these molecules, but the authors do seek to comment on the nature of **3b** and **3e** by analysis of the X-ray structures, and a calculation would add to the information that can be deduced from the structures alone.

Our responses: as mentioned earlier, the mixed ylide characters referred to chemical reactivity, not X-ray features. The reactivity of the I,S-ylide was analogous to I,P-ylides reported by Moriarty *et. al.* (ref. 49), in which the iodine could easily leave to yield the corresponding carbene-like species. We have revised this part of the discussion to avoid confusion. As suggested by the reviewer, we carried out DFT calculations on **3a** and **3e**. The central carbon was negatively charged and both S and I were positively charged. These data

indicated a charge-separated state within this region and supported the “mixed iodonium-sulfonium ylide character” of this type of reagent. These data have been added to the revised SI.

NBO charge

S = 1.042
I = 0.776
C = -0.698
O = -0.526

COO = -1.123

Bond Length

$$b1_{(C-I)} = 2.113$$

$$b2_{(C-S)} = 1.708$$

Mayer Bond Order

$$b1_{(C-I)} = 0.902$$

$$b2_{(C-S)} = 0.967$$

NBO charge

S = 1.052
I = 0.724
C = -0.591
O = -0.491

BF₄ = -0.776

Bond Length

$$b1_{(C-I)} = 2.056$$

$$b2_{(C-S)} = 1.724$$

Mayer Bond Order

$$b1_{(C-I)} = 1.003$$

$$b2_{(C-S)} = 0.943$$

REVIEWERS' COMMENTS

Reviewer #1 (Remarks to the Author):

The authors have made the corresponding revisions to the original manuscript. The concerns raised by this reviewer have been finely addressed, thus I recommend its acceptance in its current form.

Reviewer #2 (Remarks to the Author):

In short, the authors have addressed the concerns expressed in my review of the first draft of their manuscript. Notably, the control reactions added to the revised manuscript are far more convincing and make the proposed mechanism plausible and clearer. This improves the manuscript considerably and widens the possible impact of this work on the field of ylide chemistry.

Overall, I recommend the publication of this fine work.

Reviewer #3 (Remarks to the Author):

The authors have satisfied most of the points in the original review.

Suitable changes have been made to the refinement of 9e.

Calculations have been added for 3a and 3e, which I think are helpful.

Standard uncertainties have been added to most distances quoted in the text. There is one obvious value remaining where the su can quickly be added (C2-O1 = 1.248...). And there are some longer distances quoted for intermolecular contacts, which really should have uncertainties too. Because these atoms aren't connected, they won't be output by default by the refinement program. They need to be requested in the refinement instructions. I'm not trying to make this more difficult than it needs to be, so I'll leave it up to the authors. As a crystallographer, I would expect to see su's on these values.

I'm afraid that I still have to disagree with the statement "...and a small elongation of the C=O bond..." for 3b. The response from the authors says that this statement is being made in comparison with the structure of p-aminoacetophenone. The CSD refcode for that crystal structure is AMACPH. Measuring the C=O bond in that structure gives the value 1.215 as the authors suggest. BUT, that value comes with a standard uncertainty. The original publication shows 1.215(5). Comparing this to 1.230(4) in 3b, we find that the difference is 0.015 and 3 x the standard uncertainty of the difference is 0.019. This means that comparison to AMACPH does not support the statement concerning the elongation of the bond. Moreover, the CSD contains a newer determination of the same p-aminoacetophenone crystal structure, with refcode AMACPH01. In that structure, the C=O bond is found to be 1.230 angstrom (the uncertainty is irrelevant because the difference is zero). So elongation of the C=O bond in 3b compared to that in p-aminoacetophenone is not supported by the crystallographic evidence.

Response to reviewers:

Reviewer #1 (Remarks to the Author):

The authors have made the corresponding revisions to the original manuscript. The concerns raised by this reviewer have been finely addressed, thus I recommend its acceptance in its current form.

Our responses: We greatly appreciate the reviewer's recommendation of our work for publication in *Nature Communications*.

Reviewer #2 (Remarks to the Author):

In short, the authors have addressed the concerns expressed in my review of the first draft of their manuscript. Notably, the control reactions added to the revised manuscript are far more convincing and make the proposed mechanism plausible and clearer. This improves the manuscript considerably and widens the possible impact of this work on the field of ylide chemistry.

Overall, I recommend the publication of this fine work.

Our responses: We greatly appreciate the reviewer's recommendation of our work for publication in *Nature Communications*.

Reviewer #3 (Remarks to the Author):

The authors have satisfied most of the points in the original review.

Suitable changes have been made to the refinement of 9e.

Calculations have been added for 3a and 3e, which I think are helpful.

Our responses: We thank reviewer 3 for the helpful suggestions and kind words on our revised work.

Standard uncertainties have been added to most distances quoted in the text. There is one obvious value remaining where the su can quickly be added (C2-O1 = 1.248...). And there are some longer distances quoted for intermolecular contacts, which really should have uncertainties too. Because these atoms aren't connected, they won't be output by default by the refinement program. They need to be requested in the refinement instructions. I'm not trying to

make this more difficult than it needs to be, so I'll leave it up to the authors. As a crystallographer, I would expect to see su's on these values.

Our responses: According to reviewer 3's suggestion, standard uncertainty data of two longer intermolecular contacts (I1-F2 and S1-F2) in **3e** have been calculated and added in the revised manuscript (I1...F2:2.965(3) Å, S1...F2:3.802(3) Å). The original data could be seen in the following screenshot picture of .lst files.

```
L.S. 10
PLAN 1
SIZE 0.04 0.06 0.06
TEMP -100
CONF
RTAB dist I1 F2
MORE -1
BOND $H
fmap 2
acta
OMIT 5 11 -8
WGHT 0.0251 0.1781
FVAR 3.81863
REM <olex2.extras>
REM <HklSrc "%.\zfwang8_173k.hkl">
REM </olex2.extras>
```

```
Distance DIST
2.9650 (0.0026) I1 - F2
```

```
L.S. 10
PLAN 1
SIZE 0.04 0.06 0.06
TEMP -100
CONF
RTAB dist S1 F2
MORE -1
BOND $H
fmap 2
acta
OMIT 5 11 -8
WGHT 0.0251 0.1781
FVAR 3.81863
REM <olex2.extras>
REM <HklSrc "%.\zfwang8_173k.hkl">
REM </olex2.extras>
```

```
Distance DIST
3.8017 (0.0028) S1 - F2
```

Since we removed the discussion of C=O bond length in the revised manuscript, the data (C2-O1 = 1.248(5) Å) in main text has been deleted accordingly (remained in legend of figure 2).

I'm afraid that I still have to disagree with the statement "...and a small elongation of the C=O bond..." for 3b. The response from the authors says that this statement is being made in comparison with the structure of p-aminoacetophenone. The CSD refcode for that crystal structure is AMACPH. Measuring the C=O bond in that structure gives the value 1.215 as the authors suggest. BUT, that value comes with a standard uncertainty. The original publication shows 1.215(5). Comparing this to 1.230(4) in 3b, we find that the difference is 0.015 and 3 x the standard uncertainty of the difference is 0.019. This means that comparison to AMACPH does not support the statement concerning the elongation of the bond. Moreover, the CSD contains a newer determination of the same p-aminoacetophenone crystal structure, with refcode AMACPH01. In that structure, the C=O bond is found to be 1.230 angstrom (the uncertainty is irrelevant because the difference is zero). So elongation of the C=O bond in 3b compared to that in p-aminoacetophenone is not supported by the crystallographic evidence.

Our responses: We agree with reviewer 3's argument that the elongation of C=O is ambiguous, and we have removed this part of discussions. We would like to thank reviewer 3 for his/her professional advice on interpreting X-ray details.